# Hyperspectral Imaging for Skin Feature Detection: Advances in Markerless Tracking for Spine Surgery

**Francesca Manni** [1,*] , **Fons van der Sommen** [1] , **Svitlana Zinger** [1] , **Caifeng Shan** [2] , **Ronald Holthuizen** [3] , **Marco Lai** [4] , **Gustav Buström** [5] , **Richelle J. M. Hoveling** [6] , **Erik Edström** [5] , **Adrian Elmi-Terander** [5] **and Peter H. N. de With** [1]

[1] Department of Electrical Engineering, Eindhoven University of Technology, 5600 MB Eindhoven, The Netherlands; fvdsommen@tue.nl (F.v.d.S.); s.zinger@tue.nl (S.Z.); P.H.N.de.With@tue.nl (P.H.N.d.W.)

[2] College of Electrical Engineering and Automation, Shandong University of Science and Technology, Qingdao 266590, China; Caifeng.Shan@gmail.com

[3] Philips Healthcare, 5684 PC Best, The Netherlands; ronald.holthuizen@philips.com

[4] Philips Research, 5656 AE, HTC 34 Eindhoven, The Netherlands; marco.lai@philips.com

[5] Department of Neurosurgery, Karolinska University Hospital and Department of Clinical Neuroscience, Karolinska Institutet, SE-171 46 Stockholm, Sweden; gustav.burstrom@ki.se (G.B.); erik.edstrom@sll.se (E.E.); adrian.elmi-terander@sll.se (A.E.-T.)

[6] Quest Medical Imaging BV, 1775 PW Middenmeer, The Netherlands; richelle.hoveling@quest-innovations.com

* Correspondence: f.manni@tue.nl



**Featured Application: Current spinal navigation systems rely on optical reference markers to detect the patient's position. To bypass the use of current reference marker solutions, our work aims at a reliable and simple skin feature detection technology for improving clinical workflow during spine surgery. Unfortunately, reference markers or reference frames can be displaced or obscured during the surgical procedures. We present a solution by applying hyperspectral imaging (HSI) to directly detect skin features for navigation. The initial results demonstrate that HSI has the potential to replace marker-based solutions and can serve as a platform for the further development of markerless tracking.**

**Abstract:** In spinal surgery, surgical navigation is an essential tool for safe intervention, including the placement of pedicle screws without injury to nerves and blood vessels. Commercially available systems typically rely on the tracking of a dynamic reference frame attached to the spine of the patient. However, the reference frame can be dislodged or obscured during the surgical procedure, resulting in loss of navigation. Hyperspectral imaging (HSI) captures a large number of spectral information bands across the electromagnetic spectrum, providing image information unseen by the human eye. We aim to exploit HSI to detect skin features in a novel methodology to track patient position in navigated spinal surgery. In our approach, we adopt two local feature detection methods, namely a conventional handcrafted local feature and a deep learning-based feature detection method, which are compared to estimate the feature displacement between different frames due to motion. To demonstrate the ability of the system in tracking skin features, we acquire hyperspectral images of the skin of 17 healthy volunteers. Deep-learned skin features are detected and localized with an average error of only 0.25 mm, outperforming the handcrafted local features with respect to the ground truth based on the use of optical markers.

**Keywords:** hyperspectral imaging; feature detection; spine surgery; markerless tracking; deep local features

---

## 1. Introduction

In spinal fixation surgery, navigation systems provide accuracy to allow the safe placement of pedicle screws [1,2]. Compared to free-hand techniques, which rely solely on the expertise of the surgeon, a safer pedicle screw placement is achieved and the risk of damaging neurovascular structures is reduced [3–5]. The improved accuracy afforded by surgical navigation also facilitates minimally invasive surgery (MIS), which strives to reduce tissue trauma and blood loss, shortening recovery time [6]. After the initial co-registration of patient position and imaging data, accuracy in navigated spinal surgery relies on the continuous tracking of the patient position. Thus, tracking should be sustained without loss throughout the whole procedure and should also be able to compensate for patient movements during breathing and surgical manipulation. The current, commercially available spinal navigation systems utilize markers or dynamic reference frames (DRF) to track the patient's position [7,8]; however, markers or DRFs can be dislodged or obscured during the surgical procedure, resulting in the loss of navigational feedback and accuracy. In addition, DRFs are reportedly bulky and require adequate anchoring to the spine for best accuracy.

We have previously reported on an augmented reality surgical navigation system (ARSN) using adhesive optical markers placed on the skin and detected by live video cameras for tracking [9,10]. Since patient movements during surgery are typically well controlled and mainly reflect intentional movements of the surgical table, the relations between the skin surface and the deeply located spine structures are maintained. Thus, navigational accuracy can be achieved using surface-based tracking solutions [11]. The ARSN system using adhesive skin markers for patient position tracking has previously been extensively investigated in cadaveric and clinical studies and has demonstrated high accuracy [6,9,10,12–15]. Although easily detected and unobtrusive, adhesive optical markers still require placement within the surgical field, and a number of such markers need to be visible for tracking to be maintained. In contrast to marker-based solutions with single fixation points, typically occurring skin features such as spots, moles or changes in pigmentation are more uniformly distributed and can potentially provide accurate tracking without the shortcomings of other solutions. To explore the potential of skin feature detection, we have employed hyperspectral imaging (HSI). HSI is an optical imaging technique that captures a wide range of the electromagnetic spectrum, making it suitable for the detection of skin features even beyond the visible spectrum. HSI is non-ionizing and non-invasive and relies on the use of special cameras to acquire two-dimensional images across adjacent narrower spectral bands, reconstructing the reflectance spectrum for every pixel [16]. Compared with a normal RGB (Red Green Blue) camera, the spectral information is enriched by acquiring images over a larger spectral range with a narrower spectral band. This allows HSI to capture features which are not detectable with the visible wavelengths and consequently to capture information below the skin surface. xw The technology is well established in the remote sensing field and has recently received attention in medical imaging. The potential of HSI has been demonstrated in the medical field by Lu et al. [17], who employed HSI as a non-invasive tool for tumor visualization and classification. Fabelo et al. [18] developed a system that allows the capture of hyperspectral (HS) images of the in-vivo brain surface during neurosurgical operations. Their system acquired two HS data cubes: one in the visible-to-near-infrared (VNIR) range covering 400–1000 nm, and another in the near-infrared (NIR) range covering 900–1700 nm. Pixel-based classification using a support vector machine (SVM), an artificial neural network (ANN) and a random forest (RF) classifier were performed and compared to distinguish between normal tissue, tumor tissue and background elements. HSI has been also used to assess tissue perfusion, the identification of blood vessels, the differentiation of arteries from veins as well as blood-flow measurement in the skin and ischemia detection [19–22]. Therefore, it can be used to obtain information about the superficial anatomical structure, allowing us to look deeper below the skin surface and capture features which are not detectable with visible wavelengths [23,24].

We have previously presented a novel approach for tissue tracking by using multispectral imaging (MSI). We have acquired multispectral (MS) images from the skin of healthy volunteers with an MS camera, which uses a filter wheel and multiple light sources [25]. We have proved the feasibility of

tracking skin features without any markers, reaching sub-pixel accuracy for wavelengths at 430 nm and 970 nm, when a simulated movement transformation is applied to the MSI images.

The aim of this study is to validate the use of HSI for skin feature detection and to propose a method for the integration of the HSI technology with an image-guided spinal navigation system to facilitate patient tracking.

We present an investigative study aiming at a future integration of the HSI technique with an image-guided spinal surgery system to improve the clinical system suitability and achieve markerless patient tracking. The contributions of our method are as follows: (1) we propose a learning network-based approach for local skin feature detection and matching, (2) we explore novel HS data based on a new experimental camera, and (3) we validate the HS study on data acquired in human volunteers.

## 2. Materials and Methodology

This section describes the HSI instrumentation used to acquire the dataset, as well as the proposed methodology. An HS camera system (Quest Medical Imaging BV, Middenmeer, The Netherlands) has been prototyped; the system consists of a snapshot camera, which is intended to acquire both spatial and spectral information with one single exposure on the area detector. Compared to the conventional spectral imaging modalities, the snapshot system used is simple and robust to motion artifacts. The system applies a white light source to provide illumination and collects the skin-reflection images using the snapshot HS camera [26].

We have developed a computer vision framework to perform skin feature detection and matching for different breathing conditions of a subject. The framework consists of preprocessing the acquired images, detecting the skin features and estimating the feature location with respect to four adhesive optical markers glued in the field of view. The glued markers represent the ground truth in this study. We first apply band selection algorithms and then show how they can outperform band extraction methods. Then, we evaluate two different approaches for skin feature detection, applying a handcrafted local feature algorithm and adapting a deep learned local feature detection algorithm to our case. Afterwards, we evaluate the system on 17 healthy subjects. The error is assessed using adhesive optical markers on the skin of the volunteers to measure the skin displacement. In order to prove and exploit the advantage of sampling throughout a wider spectral range from the visible to the near-infrared light range, the obtained results are compared with the results found when extracting features from RGB color images synthesized and transformed by the HS data cubes.

An overview of the designed framework is depicted in Figure 1.

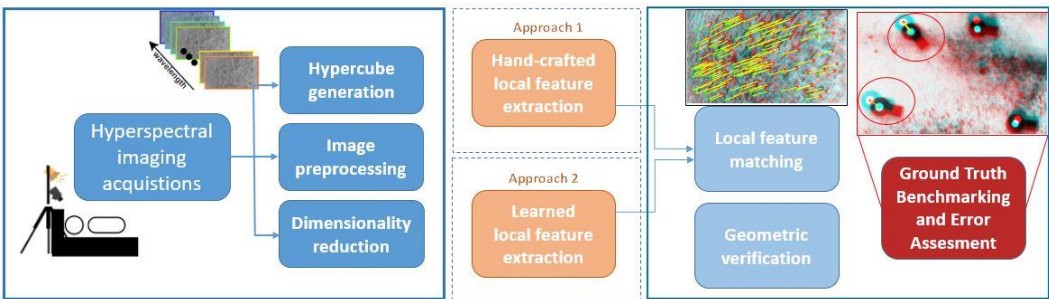

**Figure 1.** Schematic view of the complete framework. The hyperspectral (HS) cubes are preprocessed and used for feature detection and matching in order to estimate the 2D feature localization. The 2D Euclidean distances for the matched inliers are computed and compared with the Euclidean distances found with the optical markers, which represent the ground truth in this study.

### 2.1. Image Acquisitions

Images are acquired using a snapshot HSI system (Hyperea, Quest Medical Imaging B.V., Middenmeer, The Netherlands). The camera and the spectral response are visualized in Figure 2b,c. Each wavelength is covered with narrow spectral response bands to reach a high spectral resolution. The main characteristics of the camera are summarized in Table 1.

**Table 1.** Hyperspectral camera characteristics.

| Spectral Sensitivity | | Mosaic Sensor | | Images | |
|---|---|---|---|---|---|
| Spectral range | 450–950 nm | VIS (Visible) | 5 × 5 | Resolution (upscaled) | 2048 × 1080 px |
| Number of bands | 41 | NIR (Near-infrared) | 4 × 4 | Resolution spectral image | ~500 × 250 px |
| Average bandwidth | ~12 | | | Frame rate | 16 fps |

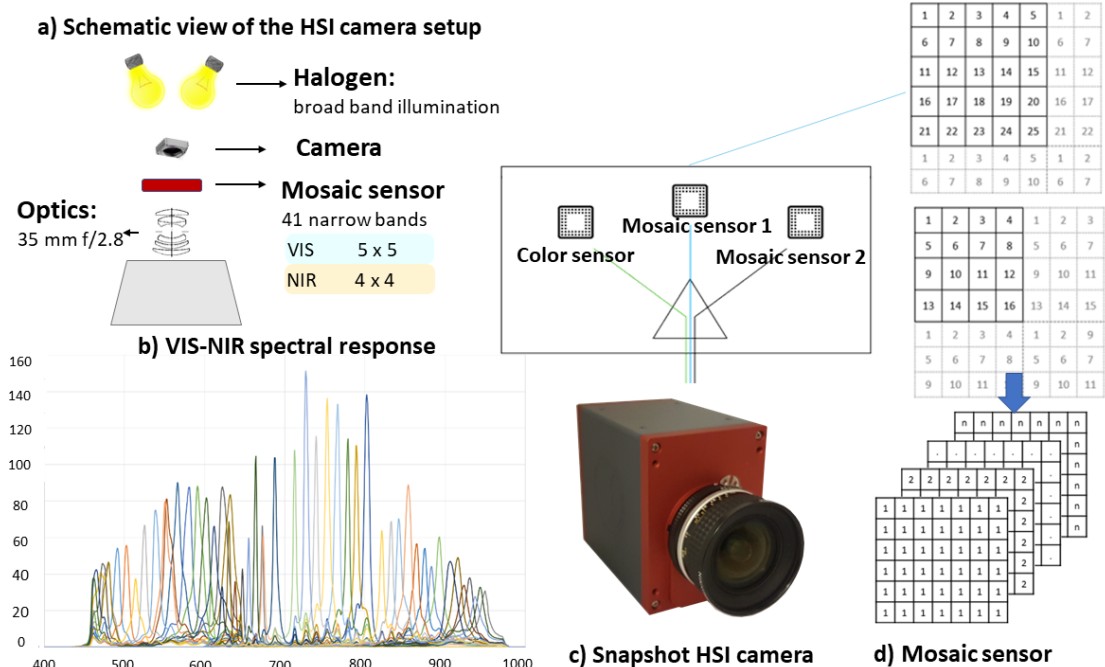

**Figure 2.** (**a**) Schematic view of the hyperspectral imaging (HSI) camera setup. (**b**) Plot of the spectral response of the camera, where the different lines correspond to the individual channel responses. Similar to an RGB camera with three filters for each color pixel, in a snapshot HS camera, each sensor is composed of several pixels that are responsive for a wavelength in the band spectrum. (**c**) Experimental prototype of the snapshot HSI camera. (**d**) Mosaic sensor technology. The picture describes a tiled-filter approach where different wavelengths are captured for different cavity heights.

The camera, running at 16 fps, captures 41 spectral bands in one snapshot, which are equally distributed in the visible near-infrared (VIS-NIR) range of 450–950 nm. The experimental set-up consists of a charge-coupled device camera (CCD) and three sensors—RGB, VIS and NIR—of which two are mosaic spectral sensors and one is a spectrally optimized lens. For 2D reference, a high-resolution color image is included with each HS dataset. The color image is co-registered with the HS data with subpixel accuracy, facilitating accurate data fusion and visualization methods. The two mosaic sensors have a 4 × 4 filter pattern for the VIS range and 5 × 5 pattern for the NIR range in order to build an HS snapshot system. Each HS image acquisition leads to a data cube with a spatial resolution of 1080 × 2048 and 41 spectral bands after processing. The HS camera is based on a tiled-filter approach (Figure 2d), where pixels are individually filtered with narrow Fabry–Pérot bandpass filters. This method allows imaging without the need to scan in either the spatial dimension (e.g., moving the

camera) or to change the pass-through characteristics of an optical filter. To uniformly illuminate the field of view, the camera was mounted on a tripod with a 500 W 240 V halogen light (Philips Lighting B.V., Eindhoven, The Netherlands) fixed 10 cm above it. In this study, the distance between the camera and the subject was 20 cm, resulting in a 15 × 15 cm field-of-view (Figure 2a). For every acquisition, the time required is approximately one second. However, in the current setup, a postprocessing step is required to construct the HS data cubes. A schematic view of the experimental setup is visualized in Figure 3.

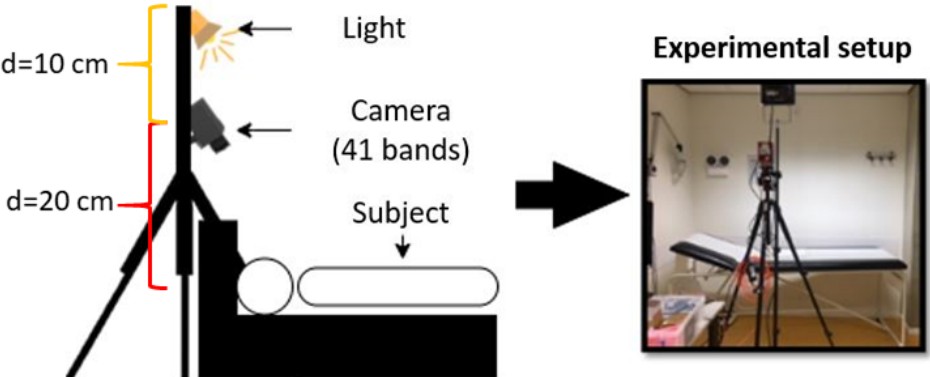

**Figure 3.** Schematic view of the hyperspectral acquisition set-up, consisting of a light source, a 41-channel hyperspectral camera and an exam table.

## 2.2. Experimental Study Protocol

A human subject study was performed in the laboratory of the Video Coding Architecture (VCA) Research Group (Eindhoven University of Technology, Eindhoven, The Netherlands). Figure 4a shows the experimental set-up view. Each subject provided their written informed consent before participating in this study, which was approved by the ethical committee (METC) of the Maxima Medical Centre (Eindhoven, The Netherlands). Seventeen healthy subjects, aged 26 to 35 years old, participated in the study. Each volunteer was asked to follow the following protocol:

1.  Inhale and hold their breath for approximately 3 s.
2.  Exhale and hold their breath for approximately 3 s.

For each subject, in every phase of the experimental protocol, one HS data cube is acquired, setting the appropriate exposure time for every sensor channel. Four adhesive optical markers in the field of view represent the ground truth in the study. In order to determine the 2D location of the markers, we detect and map the four optical markers before and after the breathing phase. Then, during the skin feature detection phase, the images are cropped and the markers are disregarded, as shown in the middle, red box in Figure 4b. The HS raw image data and white reference images are normalized to correct for the dark current influence and illumination intensity differences. To perform initial calibration, prior to every measurement, the white and black references are acquired by positioning a white reference sheet in the field of view and by keeping the camera shutter closed, respectively. It is important for the white reference sheet to cover the whole field of view to correct for the white light, as the sensor is tiled and white reference correction needs to be performed for the complete field of view on all the pixels in the sensor.

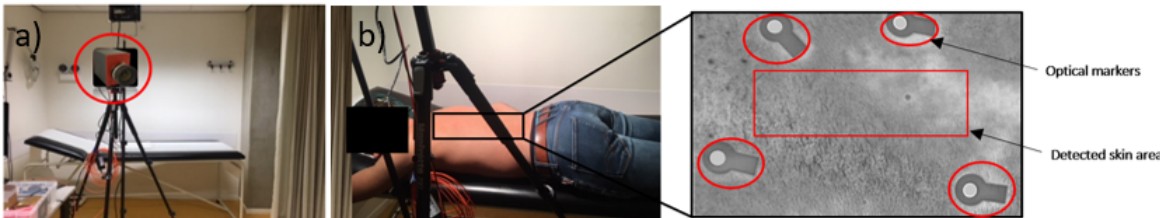

**Figure 4.** (**a**) Experimental set-up view, where the hyperspectral camera and the light source are mounted on a tripod. (**b**) The back of the subject is scanned by using optical markers as references (see the magnified view box).

### 2.3. Hyperspectral Image Processing

Preprocessing is a crucial step for HS image analysis, since HSI data suffers from high dimensionality, redundancy within image bands and instrumentation noise. First, the raw reflectance data are normalized in order to correct for the dark current noise by applying the following computation:

$$I_{ref} = \frac{I_{raw} - I_{dark}}{I_{white} - I_{dark}},\tag{1}$$

where $I_{ref}$ denotes the normalized reflectance value, $I_{raw}$ is the diffuse reflectance value, $I_{white}$ the intensity value for a white reference plate and $I_{dark}$ represents the dark reference, acquired by keeping the camera shutter closed. HS images are typically degraded due to noise and low contrast [16]. To effectively enhance the IR image, contrast methods are therefore required to enhance the global contrast and the perception of details without highlighting the noise and introducing undesirable artifacts [27]. Conventional adaptive histogram equalization is based on separate histograms for different image regions, which are used to redistribute the local intensity level values. This approach solves the main limitation of having a global histogram equalization, which tends to stretch out the intensity range of an image, resulting in the over-enhancement of the background (introduction of noise) and a contrast deterioration of small targets [27]. As an alternative, adaptive histogram equalization has been proposed to improve contrast enhancement. However, this can also visually amplify noise due to the over-enhancement effect in the homogeneous regions. Contrast-limited adaptive histogram equalization (CLAHE [28]) prevents this by limiting the amplification, and it has been successfully applied in infrared imagery and surveillance imagery. A dedicated Gaussian filter and contrast-limited adaptive filtering have been designed for the VIS and the NIR wavelength range, respectively. The Gaussian filter size is $2\sigma + 1$, where $\sigma$ represents the standard deviation, set at $\sigma = 0.05$. To enhance the contrast, we employ CLAHE, setting a normalized intensity limit of 0.05 for the VIS range and 0.1 for the NIR. The clip limit for the NIR range is higher, since the contrast within this range is lower compared to the VIS images.

For each subject, a region of interest (ROI) is manually delineated for further analysis in order to discard the optical markers positioned in the image edges. The selected ROI is chosen as the closest area to the optical markers to ensure the same displacement during breathing (Figure 4b).

### 2.4. Hyperspectral Band Selection

The redundancy of information among bands in HS images is a well-known problem, which can be attributed to the relatively high spectral resolution. HS images typically comprise a large number of bands, resulting in high-dimensional data. The high dimensionality leads to considerable computational demands and high storage costs. Therefore, to analyze the 41-dimensional HS data cubes captured by our system, we apply a dimensionality-reduction processing step. Dimensionality-reduction techniques can be divided into band extraction and band selection methods. One of the most common band extraction methods is principal component analysis (PCA), which projects the data in a low-dimensional space where the spectral information is decorrelated [29]. PCA

generates a new set of linearly uncorrelated variables, where the first few retain most of the signal variation occurring in all original variables. However, the linear projection transformation leads to the mixing of the original spectral information.

It should be noted that this mixing phenomenon poses an important issue for images with different scales or changes in spatial structures [30]. For this reason, we investigate an alternative that enables us to specifically address individual spectral bands. This alternative is found in band selection algorithms.

Band selection algorithms select a subset of bands without losing their physical meaning and have the advantage of preserving the relevant original information. Since we are interested in detecting local features in the spectral images, we have adopted the concept of Su et al. [31]. In their work, the authors defined a Saliency-Band-Based Selection algorithm (SSBS) for HS images. A saliency band is a band that is capable of characterizing multiple objects and is rare in the HS data. In order to select salient bands with high information content, the method proposes the adoption of scale-selection techniques which can characterize the objects of an image [32]. Based on this concept, the authors proposed to imitate the optimal scale-selection process. In our previous work, Speed Up Robust Features (SURF) [33] achieved the best result in detecting MS skin features when captured at different scales. For HS images in which the target may mingle with the background, it is reasonable to use a scale-selection technique and find extremum points for selecting salient bands, as suggested in [31]. We calculate the determinant of the Hessian matrix for the entire HS data cube in order to determine the local extrema for each band. A salient band is selected if it has a higher amount of local extrema compared with its neighboring bands (i.e., left and right adjacent bands).

For benchmarking, we also applied an unsupervised band selection algorithm, which was the fast Volume-Gradient-Based Band Selection (VGBS) method from Geng et al. [34] because it offers a higher accuracy in comparison with other band selection methods [31]. VGBS is a geometry-based method that identifies the bands with the maximum ellipsoid volume for capturing spectral information. The bands are treated as data points in a high-dimensional space. At the beginning of the algorithm, all bands are considered as candidate bands and constitute a parallelotope. Then, bands leading to the minimal losses of the parallelotope volume are removed iteratively, until the desired number of bands in the subset remains. VGBS can effectively reduce the correlation among bands, since low-correlated bands often represent a large-volume parallelotope [34].

### 2.5. Local Feature Detection

The third step of our proposed approach is to find local features in the spectral images, which are detected in both breathing phases, and estimate the 2D displacement to assess the feature location.

In earlier work, we demonstrated the feasibility of MSI for the improved detection of skin features after a simulated 2D rigid image shift [25]. We have analyzed MS images for 30 human subjects at eight different wavelengths, achieving high accuracy in detecting the simulated 2D applied displacement with on-body features (with an error lower than 40 μm). In addition, we have shown the feasibility of skin feature detection using handcrafted local feature algorithms, such as Speed Up Robust Feature, (SURF) [25,33] and Maximal Stable Extremal Region (MSER) [35], as possible methods which are capable of detecting veins, moles and skin patterns.

We thus apply the same concept for the processing of the HS images, detection of the skin features and the matching of those features to estimate the 2D-Euclidean distance after one-to-one-matching. For each subject, we consider a pair of HS data cubes obtained in the different phases of breathing.

To detect the desired features, it is crucial to extract informative regions or points on the skin, which should be invariant to geometric transformations and insensitive to degradation effects, such as illumination changes and intensity changes produced by noise. We first detect skin features by applying SURF as a feature detection algorithm [33,36] to obtain the best performance. We then adapt the algorithm for this study where real subject motion is included, as described in the experimental protocol (Section 2.2).

The SURF algorithm exploits integral images to build the scale space. This is an approximation of the Gaussian scale space that reduces the computation time. Regarding the descriptor, it is formed only by 64 dimensions, and the orientation is estimated using the Haar wavelet [37].

As an alternative to handcrafted local descriptors, convolutional descriptors extracted from convolutional neural networks (CNNs) have recently shown promising results [38,39]. A new architecture has been proposed that learns and extracts multi-scale, local descriptors, called deep local features (DELF) [39]. Compared to conventional handcrafted local feature detection algorithms, the DELF framework offers a more accurate feature matching and geometric verification through an attention-based mechanism that learns which features are most discriminative. The DELF module is designed for image retrieval and consists of the following four steps: dense localized feature extraction, key point selection, dimensionality reduction and image retrieval. To evaluate this method for our purpose, we execute it in parallel to the conventional SURF approach. The learned feature extraction process can be divided into two steps: first, dense features are generated from images with a fully convolutional network (ResNet-50), which is initially trained for the classification of ImageNet [40]. Then, the obtained feature maps are regarded as a dense grid of local descriptors. The features are localized based on their receptive fields, which can be computed by considering the configuration of convolutional and pooling layers of the fully convolutional network (FCN). The pixel coordinates of the center of the receptive field are used as the feature location. Second, an attention score function is trained to assess the relevance of the extracted features. The algorithm handles different image scales by constructing an image pyramid and applying an FCN to each level, meaning that is possible to obtain features that describe an image region at different scales. As in the original approach, we scale our images with the following coefficients: 0.25, 0.3536, 0.5, 0.7071, 1.0, 1.4142, 2.0. In the dimensionality-reduction step, the feature dimension is reduced by PCA. The features produced are vectors of length 1024. Each DELF descriptor includes its location in the image, the score and the scale at which the feature was found. Then, we extract the descriptors and their location for every pair of images, using the pre-trained DELF network with landmark images and we apply the K-dimensional tree to find K nearest neighbors for each descriptor. The network architecture for feature extraction and matching is depicted in Figure 5.

*2.6. One-to-One Matching*

Feature matching is established based on the pairwise distance between two extracted feature descriptors during the two breathing phases, which are formally specified by

$$F_{bi} = [f_{bi_1}, f_{bi_2}, ..., f_{bi_n}] \quad \text{and} \quad F_{bo} = [f_{bo_1}, f_{bo_2}, ..., f_{bo_n}], \tag{2}$$

where $F_{bi}$ is the feature vector extracted before the movement of the back of the subject and $F_{bo}$ is the vector extracted after the movement of the back of the subject.

Lastly, a geometric transformation (as indicated in the diagram in Figure 5) is established using random sample consensus (RANSAC) [41], which removes outliers to make matching more consistent. In our case, the purpose of RANSAC is to estimate transformations from random subsets of matched feature pairs. These transformations are then tested against the full set of matches, and the final transformation is the one that aligns the highest number of candidate matched feature pairs.

For benchmarking, we compare the markerless approach with the optical-marker based methodology. The adhesive optical markers, representing the ground truth, are glued at the back of the subject and easily detected and matched with the described approaches. The marker detection will serve as a reference to estimate the error made by using skin feature detection algorithms (SURF and DELF), as shown in Figure 1.

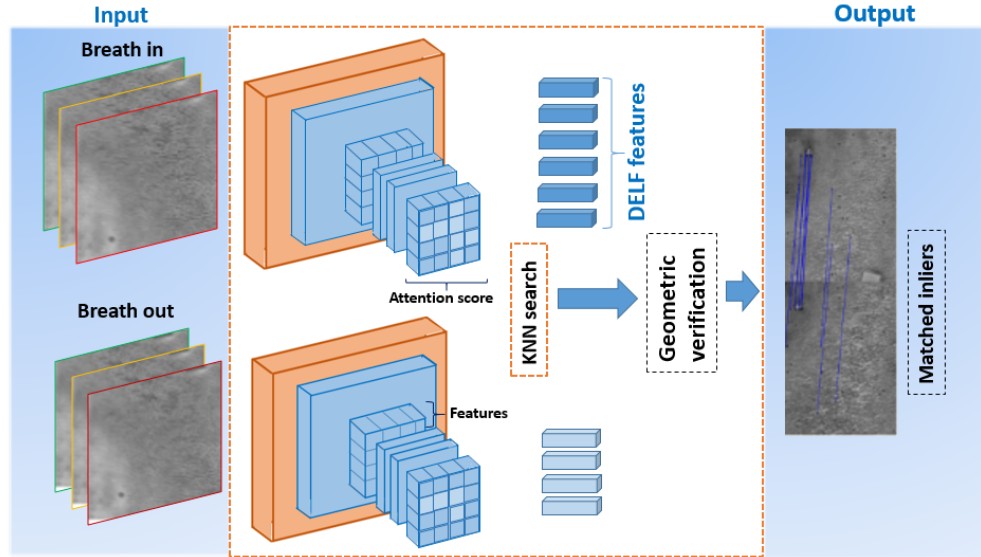

**Figure 5.** Architectures for feature extraction and matching adapted from Noh et al. [39]. The left panel shows the input HS data corresponding to the two different breathing phases. The middle panel depicts the feature extraction, indicating the attention scores assigned to relevant features and used for feature selection. The matched features are shown as the final output of the system in the right panel. KNN: K nearest neigbor.

## 2.7. 2D Feature Localization

After the 2D geometric verification, for each matched pair of features, we compute the Euclidean distance $D(x_{in}, x_{out}, y_{in}, y_{out})$ between the two different breathing phases, which results in the following:

$$D(x_{in}, x_{out}, y_{in}, y_{out}) = \sqrt{(x_{in} - x_{out})^2 + (y_{in} - y_{out})^2}, \qquad (3)$$

where $x_{in}$ and $y_{in}$ are the 2D coordinates of a single matched feature in the first phase of the protocol and $x_{out}$ and $y_{out}$ are the 2D coordinates for the same feature in the second phase. The mean distance is computed for both SURF and DELF extracted features and used for performance comparison. The benchmarking of the 2D Euclidean distance between the breathing phases for each detected adhesive optical marker is calculated in the same way as shown in Equation (3).

## 2.8. RGB Benchmark Experiment

The advantage of HSI compared to standard RGB is deeper tissue penetration and wider spectral coverage. Aiming at a comparison with standard RGB, we synthesize and transform the HS radiance data to RGB color images. The color images are then used to detect skin features. In order to render and visualize HS images in a uniform RGB color space, we adopt the approach of Foster et al. [42]. The methodology introduces image transformation to the application of color metric representations and color rendering. A color space for rendering is needed to visualize a spectral image [42,43]. With the RGB color space, the radiance values are converted to CIE XYZtristimulus values $X, Y, Z$ [43]. Then, the RGB values are obtained by a linear transformation formulated by

$$\begin{bmatrix} R \\ G \\ B \end{bmatrix} = \begin{bmatrix} 3.2406 & -1.5372 & -0.4986 \\ -0.9689 & 1.8758 & 0.0415 \\ 0.0557 & -0.2040 & 1.0570 \end{bmatrix} \begin{bmatrix} X \\ Y \\ Z \end{bmatrix}. \qquad (4)$$

Figure 6 shows an example RGB rendering of an HS image. As suggested by Le Moan et al. [44], to improve appearance, the RGB levels of the image can be clipped to the level of a less specular—but still bright—region and then scaled [45]. The result is shown in Figure 6. Clipping can lead to artifacts

(such as specular glare), which can be reduced by adopting a further algorithm that preserves the relationship between different colors [46]. The clipped image is used to detect and match the 2D skin features with the method described in Section 2.5. The results are then evaluated and compared with the results found by using the HS images.

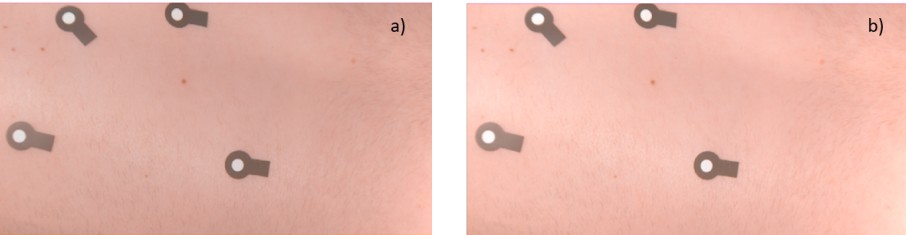

**Figure 6.** (**a**) RGB (Red Green Blue) color image obtained from the hyperspectral radiance image of subject no. 1. (**b**) a clipped version of image (**a**) with the clip level taken from the brightest region.

## 3. Experiments and Results

This section will present the results obtained with the proposed processing framework, as well as the overall results by applying the band selection algorithms. The experiments also include a comparison with the results found employing RGB images, transformed from the HS radiance data.

### 3.1. Image Preprocessing and Feature Detection

The proposed system is evaluated by assessing the 2D error made by estimating the 2D feature location using handcrafted and learned features. Thirty-four hyperpectral data cubes, from 17 healthy volunteers, were acquired using the snapshot HS camera. For each subject, the protocol described in Section 2.2 was applied, consisting of the acquisition of two hyperpectral data cubes corresponding to the inhaling and exhaling phases. For each data cube, the dimensionality reduction step was applied and the features were detected by using the described approaches, such as PCA, VGBS and SSBS. Lastly, feature matching was performed to estimate the 2D feature location after breathing. In spinal surgery applications, breathing movements should be compensated. The distances when the optical markers are detected are computed and the errors made when the skin features are employed are calculated and expressed in millimeters (mm).

SURF and DELF are used as feature detection and extraction algorithms. A pre-trained DELF network for learned feature extraction is deployed. The network is trained with landmark images which are different from HS images. However, these experiments, using features trained on different images, merely serve as a preliminary test of the system. When SURF is selected as feature detection algorithm, a feature threshold equal to 500—two octaves and four scale levels per octave—is employed.

### 3.2. Band Extraction and Selection Methods

In order to further illustrate the importance of selecting the most meaningful bands, the accuracy when the PCA, VGBS and the SSBS approaches are employed is reported in Table 2. PCA reduces the data into the most important wavelengths that account for the largest variance in the dataset. Using eigenvalue decomposition of the covariance matrix of the HS data cube, the image that corresponds to the first, second, and third largest eigenvalue are selected as the first three principal components. Only these components are selected for further processing, because they contain the largest variance of information.

The VGBS algorithm was evaluated for different numbers of bands (5, 10, 15), but only the results obtained with 10 bands have been reported, because they were found to be the most competitive. When 10 bands are selected, the mean error for estimating the 2D displacement is 0.42 ($\pm$0.05) mm when SURF is used as a feature detector algorithm, while an error of 0.27 ($\pm$0.06) mm is found when DELF is employed instead. When 5 and 15 bands are selected, the error increases by 3% and 1%,

respectively, when SURF is applied, and an error growth of 3% is found with DELF. The plots provided in Figure A1 in the Appendix A show the detailed results obtained with the VGBS algorithm.

For the SSBS algorithm, the output number of the selected bands is determined by the input HS image itself [31] and selected bands cannot be ranked. Due to the similarity in the nature of the input images, the number of selected bands using the SSBS method is equal to 16, 15 and 14 for five, six and four subjects, respectively.

As an example, Figure 7 illustrates the selected bands for subject no. 8 when the SSBS algorithm is used. Figure 8a,b show the matching results when PCA is used as a band extraction algorithm. Alternatively, Figure 8c,d portrays the results when the band selection algorithms are used. The matched features, which are detected using DELF, are depicted in red, and those detected by SURF are shown in green. After performing the geometric verification with RANSAC to ensure sufficient inliers, the region found after the breathing of the subject is drawn in red, as in Figure 8a. As visualized in Figure 8a–d, a higher amount of inliers is found when DELF is employed for both band extraction and selection.

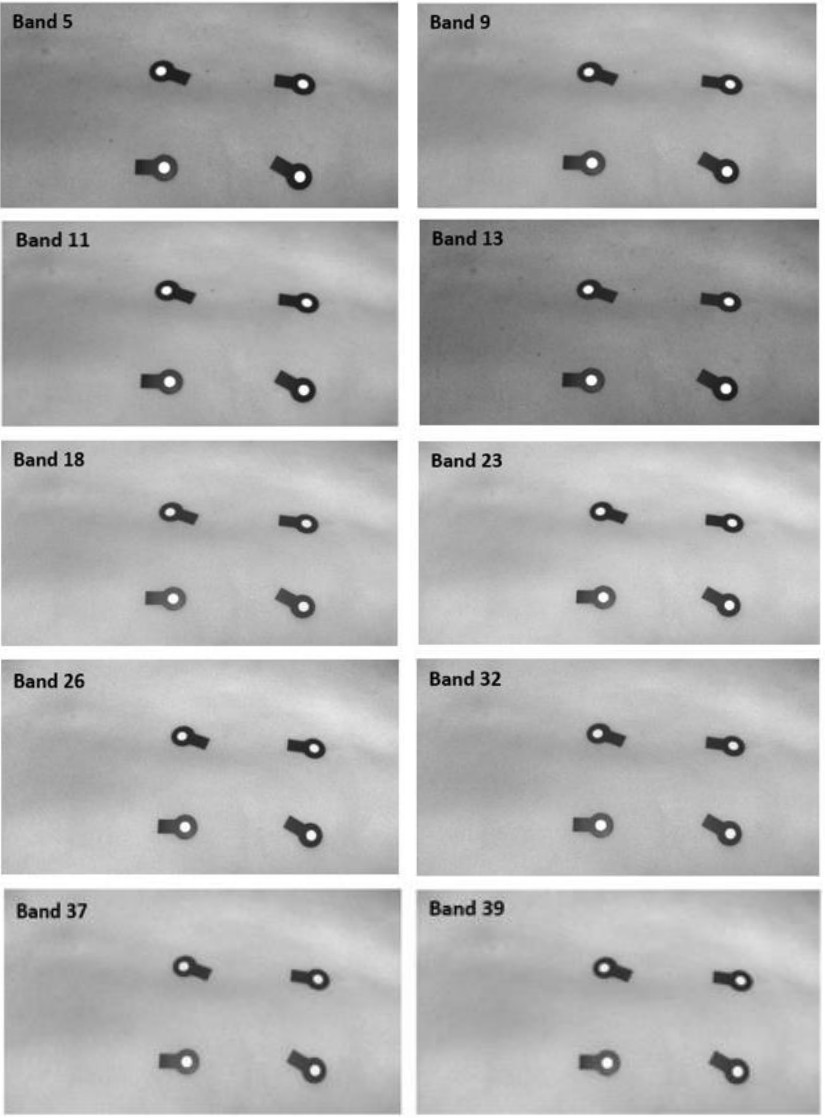

**Figure 7.** Ten selected bands are illustrated for subject no. 8. The bands are selected by using the Saliency-Band-Based Selection (SSBS) algorithm [31] adapted to our dataset.

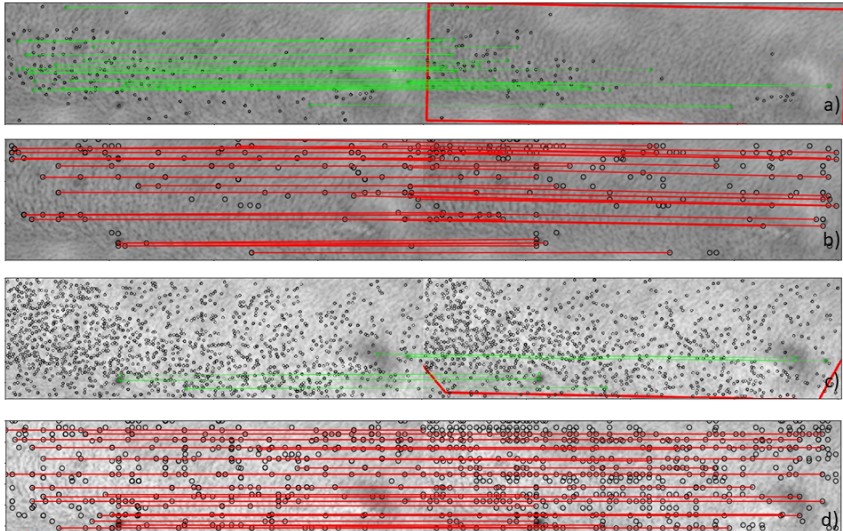

**Figure 8.** (**a**) Speed Up Robust Features (SURF) correspondences (green) after random sample consensus (RANSAC) geometric verification for subject no. 3 when principal components analysis (PCA) is used as a band extraction method. The two images correspond to the acquired images during inhalation and exhalation for the second principal component. (**b**) Deep local features (DELF) correspondences (red) after RANSAC geometric verification for subject no. 3 when PCA is used as a band extraction method. The two images correspond to the acquired images during inhalation and exhalation for the second principal component. (**c**) SURF correspondences after RANSAC geometric verification for subject no. 3 ($\approx$590 nm) when the band selection algorithms Volume-Gradient-Based Band Selection (VGBS) and SSBS are used. (**d**) DELF correspondences after RANSAC geometric verification for subject no. 3 ($\approx$590 nm) when the band selection algorithms VGBS and SSBS are used.

### 3.3. Two-Dimensional Localization Error

The average error based on the calculation of the 2D Euclidean distances between the matched feature locations is reported in Table 2, presenting the performance of the deep local feature approach and the SURF method, as described in Section 2.5.

The 2D error average per subject is calculated when three different band extraction/selection algorithms are used. Overall, a mean error of 0.39 ($\pm$0.08), 0.27 ($\pm$0.07) and 0.25 ($\pm$0.05) mm is found when deep features are detected using PCA, VGBS and SSBS, respectively. A mean error of 0.45 ($\pm$0.07), 0.42 ($\pm$0.07), and 0.30 ($\pm$0.08) mm is found when handcrafted features are detected using PCA, VGBS and SSBS methods, respectively. We observe a difference in error of slightly more than 0.10 mm when using the band selection algorithms compared to PCA for both feature extraction algorithms (SURF and DELF). It is noticed that DELF is capable of detecting skin features, which are clearly more visible when using the original bands, while the information is partially lost when PCA is used as dimensionality reduction algorithm. Overall, the trend in the results shows better performances when DELF is used in combination with the SSBS method.

The analysis of the number of inliers has been reported in Figure 9, revealing a higher amount of matched features for the majority of subjects when DELF is used, to detect and match features after outlier removal by using RANSAC.

**Table 2.** Results of the mean error per subject (in mm) in 2D skin feature localization. Handcrafted local features are compared with deep local features when PCA (three principal components) and band selection algorithms are used (10 selected bands).

|  | Handcrafted Features (SURF) | | | Deep Local Features (DELF) | | |
|---|---|---|---|---|---|---|
|  | **PCA** | **VGBS** | **SSBS** | **PCA** | **VGBS** | **SSBS** |
| Subject 1 | 0.40 | 0.39 | 0.25 | 0.38 | 0.25 | 0.22 |
| Subject 2 | 0.41 | 0.39 | 0.21 | 0.46 | 0.25 | 0.21 |
| Subject 3 | 0.44 | 0.38 | 0.20 | 0.32 | 0.24 | 0.24 |
| Subject 4 | 0.52 | 0.42 | 0.34 | 0.47 | 0.40 | 0.32 |
| Subject 5 | 0.60 | 0.41 | 0.28 | 0.46 | 0.29 | 0.35 |
| Subject 6 | 0.57 | 0.42 | 0.30 | 0.30 | 0.28 | 0.20 |
| Subject 7 | 0.46 | 0.41 | 0.28 | 0.35 | 0.25 | 0.15 |
| Subject 8 | 0.50 | 0.44 | 0.30 | 0.40 | 0.26 | 0.19 |
| Subject 9 | 0.44 | 0.43 | 0.34 | 0.40 | 0.27 | 0.23 |
| Subject 10 | 0.47 | 0.41 | 0.31 | 0.60 | 0.24 | 0.28 |
| Subject 11 | 0.41 | 0.44 | 0.23 | 0.37 | 0.24 | 0.25 |
| Subject 12 | 0.47 | 0.48 | 0.45 | 0.30 | 0.27 | 0.19 |
| Subject 13 | 0.34 | 0.44 | 0.34 | 0.46 | 0.27 | 0.31 |
| Subject 14 | 0.47 | 0.43 | 0.20 | 0.35 | 0.30 | 0.28 |
| Subject 15 | 0.34 | 0.45 | 0.20 | 0.37 | 0.28 | 0.20 |
| Subject 16 | 0.45 | 0.44 | 0.44 | 0.40 | 0.31 | 0.25 |
| Subject 17 | 0.44 | 0.38 | 0.41 | 0.31 | 0.32 | 0.27 |

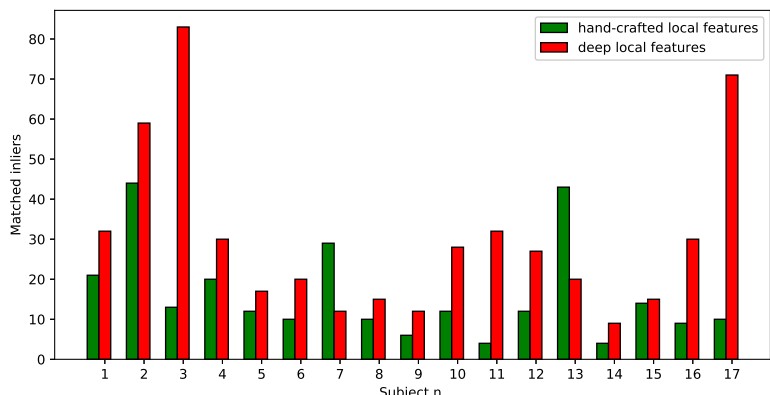

**Figure 9.** The barplot presents the average number of matched inliers per subject when the band selection algorithms are applied. The number of matched inliers using SURF and DELF are indicated and visualized in the plots above.

### 3.4. RGB Benchmark

In the last experiments, the spectral features with the features extracted from RGB-transformed radiance images are benchmarked. The RGB radiance data are transformed from the HS data cubes as described in Section 2.8. For each subject, the transformed data (Figure 6) are employed to detect the features during the different breathing phases and to estimate the 2D feature locations.

The results of the RGB-transformed color images are illustrated in Table 3. The amount of matched inliers used for geometric verification was not sufficient to perform robust feature matching (number of inliers < 2). An error larger than 1 mm is found in four cases. The algorithm failed when applied to subjects 7, 8, 9 and 12. This means that, in the geometric verification step, we could not find matched features. Apart from the noise introduced by the RGB transformation, it is easier to track skin features in the spectral images, which are enhanced when light interacts with deeper tissue layers. It is clear that HSI is much more powerful in identifying skin surface features than conventional RGB imaging.

Figure 10 illustrates an example of feature matching when the rendered RGB image is used for subject no. 3.

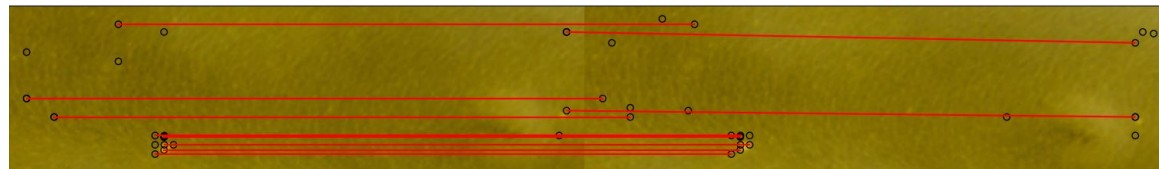

**Figure 10.** DELF correspondences after RANSAC geometric verification for subject no. 3 in the RGB-transformed image.

**Table 3.** Results of 2D mean error per subject for all the detected skin features using RGB-transformed images. For a better comparison, the results found when HSI images are used and the DELF algorithm is applied after the SSBS band selection are shown.

| Subject No. | 1 | 2 | 3 | 4 | 5 | 6 | 7 | 8 | 9 | 10 | 11 | 12 | 13 | 14 | 15 | 16 | 17 |
|---|---|---|---|---|---|---|---|---|---|---|---|---|---|---|---|---|---|
| Error (RGB–SURF) (mm) | 0.76 | 0.40 | 0.32 | 0.67 | 0.23 | n.a.[a] | n.a.[a] | n.a.[a] | 0.89 | 1.35 | n.a.[a] | 1.69 | 1.68 | 1.29 | 0.86 | 0.97 | 0.65 |
| Error (RGB–DELF) (mm) | 0.96 | 0.91 | 0.96 | 0.67 | 0.20 | n.a.[a] | n.a.[a] | n.a.[a] | 0.73 | n.a.[a] | n.a.[a] | 1.24 | 1.51 | 1.35 | 0.48 | 1.55 | 1.04 |
| Error (HSI–DELF–SSBS) (mm) | 0.22 | 0.21 | 0.24 | 0.32 | 0.35 | 0.20 | 0.15 | 0.19 | 0.23 | 0.28 | 0.25 | 0.19 | 0.31 | 0.28 | 0.20 | 0.25 | 0.28 |

[a] Not applicable. The algorithm failed when applied to subjects 7, 8, 9 and 12, identified as n.a. in the Table.

## 4. Discussion

HS images allow the acquisition of a large number of bands with spectral information, which is collected for each pixel in the image. The HSI technology scans beyond the skin surface, revealing what is not visible with the human eye. This can be beneficial for surgical guidance, when accurate and continuous patient tracking is needed. Therefore, HSI can offer an attractive solution for noninvasive skin feature detection and patient position tracking.

The currently available surgical navigation systems use either invasive or noninvasive reference markers or dynamic reference frames for patient position tracking. However, tracking can be compromised if the markers are covered or displaced during surgery, resulting in navigational inaccuracies; thus, using inherent skin features instead of markers can offer a potential improvement of patient position tracking and robustness for surgical navigation.

In this work, we have presented a human-subject study, where HSI was used to detect and match features while the subject was breathing normally. The 2D displacements between matched skin features were measured and benchmarked with adhesive optical markers. A mean error of 0.25 mm over all subjects was found when deep local features (DELF) were detected, while the system performance decreased when SURF was used as a handcrafted feature detection algorithm.

The HS images involve large data sizes in the range of Gigabytes, requiring high-performance computing power to facilitate a feasible processing time [47]. To increase the feasibility of the approach, we have evaluated different algorithms (PCA, SSBS, VGBS) for the extraction of the most informative bands to perform skin feature detection and matching. We have compared the results of extracting the most meaningful bands when PCA and two different band selection algorithms—VGBS and SSBS–were used. PCA is well established as a dimensionality reduction method for HS images; however, it leads to a loss of meaningful spectral information. We found that the SSBS method, which selects the spectral bands based on the number of local extrema, gave the best results. One of the major reasons for this result is that the conceptual idea of finding extreme local points is the basis for a scale-selection algorithm which is employed to find local features in an image.

In order to demonstrate the added value of detecting features with an HS camera, we synthesized RGB color images from the HS images for comparison. HSI images were transformed into color images, after which DELF was applied to detect and match the best possible skin features. However, the 2D skin feature localization accuracy decreased.

It should be noticed that the image acquisition system does not operate in real-time. A postprocessing phase is required to create the HS data cube for each image acquisition.

However, in an in-vivo scenario, a camera setting that is able to construct the data cube during the acquisition phase to facilitate the real-time detection and usage of skin features to directly provide navigational feedback is required. Furthermore, some technical aspects should be taken into account, such as the illumination conditions, which are less controlled in the operating room. For an in-vivo study, the above-described issues should be considered. There are two major stages, which are application/acquisition and data processing. This paper focused mostly on the data processing rather than data acquisition, and we have contributed to making this processing efficient for real-time operation. The next step is to redesign the data acquisition procedure and optimize the illumination intensity, which will create the desired improvements needed for real-time clinical application.

We show a clear improvement when detecting local features in HS skin images while employing an expensive camera. An important trade-off should be considered, where employing an HS camera in a broad wavelength range (VIS-NIR) would increase the cost of the procedure. The conclusion is that more spectral information incurs higher system costs.

The penetration depth of the HSI camera used in this study has been found to be hundreds of micrometres to a maximum of 1 mm dependent on the wavelength [23], which limits its clinical applicability to the superficial layers of the target tissue, although it was suitable for the scope of this paper.

In this work, we have computed a 2D displacement of the skin features. For future work, we want to implement our system in a 3D stereo-vision scenario, where the 3D location of the skin features can be assessed and used to correct for patient motion.

## 5. Conclusions

During spine surgery, HSIs of skin features can be used for noninvasive patient positioning and tracking. Skin feature tracking offers an extension and an improvement of current tracking systems, with the aim of optimal patient motion compensation and reliable surgical guidance.

In this paper, we have performed a human subject study acquiring HS images from the back of the body. We presented a preliminary study in which we demonstrated, as a proof-of-concept, that HSI allows the detection of on-body salient feature points and the finding of reliable correspondences after skin displacements due to breathing. Optical markers were used as ground truth, and on average. we found an overall error of 0.25 mm in computing 2D feature locations when deep local features were detected, outperforming the handcrafted SURF method by slightly more than 5%.

The preliminary results are promising and represent an initial investigation into the acquisition and detection of spectral features in human volunteers with a sub-millimeter error lower than 0.5 mm. Future development should strive towards real–time 3D applications for in-vivo surgical navigation. We are of the opinion that the presented approach has strong potential to improve workflow and reliability in navigated surgery and may serve as a reference for introducing HSI in a surgical environment.

**Author Contributions:** Conceptualization, investigation, methodology, software and writing, F.M.; supervision, conceptualization, review and editing, F.v.d.S., S.Z. and R.H.; project administration, review and editing, C.S.; review and editing, G.B., E.E., A.E.-T.; resources, R.J.M.H., M.L.; supervision, review and review, P.H.N.d.W. All authors have read and agreed to the published version of the manuscript.

**Funding:** The research activity leading to the results of this paper was funded by the H2020-ECSEL Joint Undertaking under Grant Agreement No. 692470 (ASTONISH Project).

**Acknowledgments:** We acknowledge the NVIDIA Corporation for their donation of the Titan Xp GPU used for this research and Quest Medical Imaging BV for setting the camera used for the experiments.

**Conflicts of Interest:** The authors who are affiliated with clinical academic institutions (FM, FVDS, SZ, AET, GB, EE, PHNW) have no financial interests in the subject matter, materials, or equipment or with any competing materials and did not receive any payments as part of the study. The authors affiliated with Philips Healthcare (RH, CS, ML) are employees of Philips and thereby have financial interests in the subject matter. The author affiliated with Quest Medical Imaging BV (RJMH) is an employee of Quest Medical Imaging BV and thereby has financial interests in the materials and equipment. The extent of influence on the data, manuscript structure and manuscript conclusions by these authors was limited to technical knowledge and input. Authors without conflicts

of interest had full control of all data collection, data analysis and information submitted for publication and over all conclusions drawn in the manuscript. The prototype system described in this article is currently a research prototype and is not for commercial use.

## Appendix A

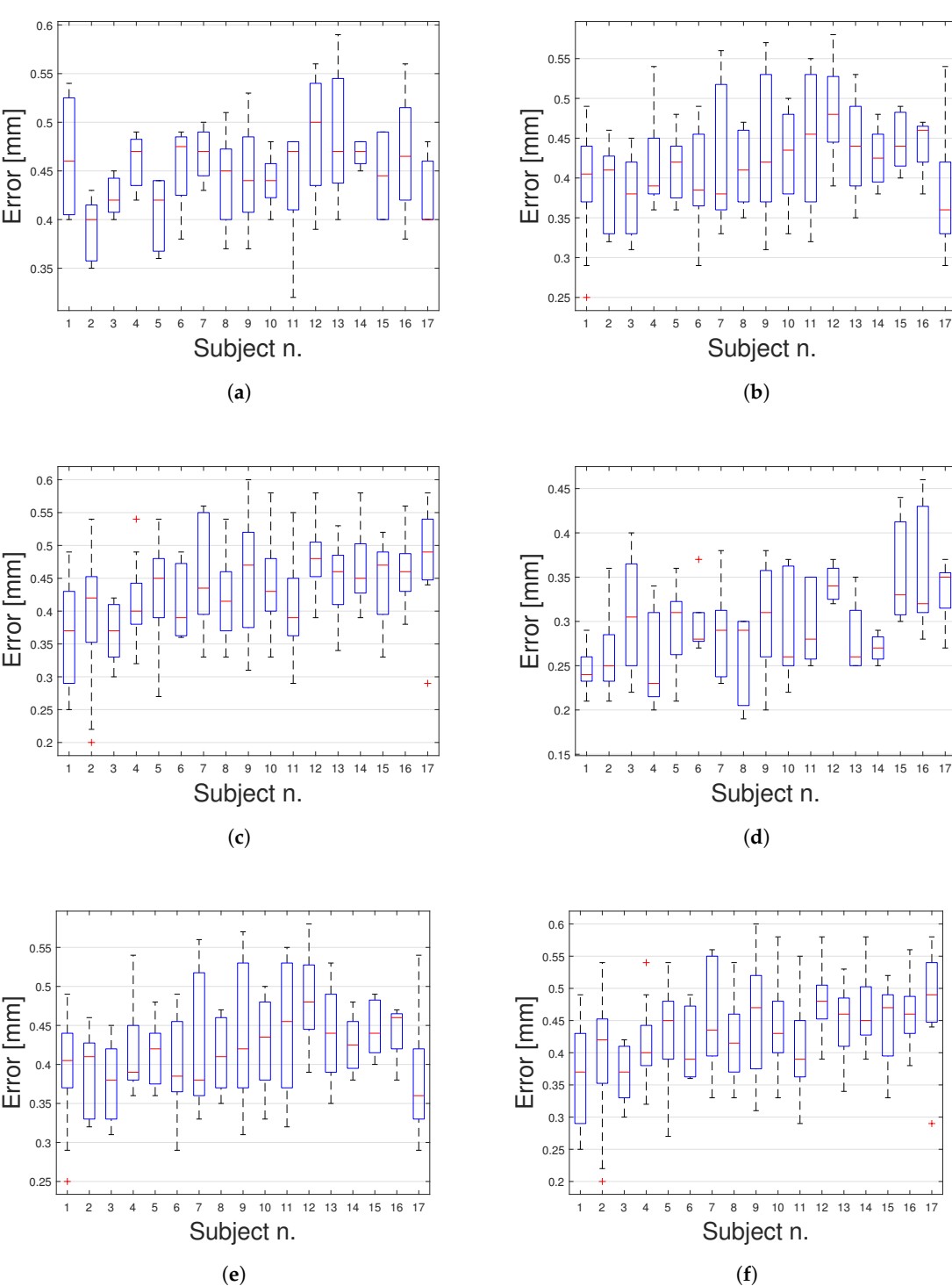

**Figure A1.** Boxplot results of the mean error for each patient, employing the VGBS algorithms with different numbers of bands. (**a**) Five selected bands using SURF. (**b**) Ten selected bands using SURF. (**c**) Fifteen selected bands using SURF. (**d**) Five selected bands using DELF. (**e**) Ten selected bands using DELF. (**f**) Fifteen selected bands using DELF.

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
