# Peer review of "Hyperspectral Imaging for Skin Feature Detection: Advances in Markerless Tracking for Spine Surgery"

_applsci, doi:10.3390/app10124078_

Round 1
Reviewer 1 Report
Manni et al. present in their manuscript a comparison between the capacities of different software algorithms in tracking skin markers for the benefit of surgery. The authors use a novel camera capable of acquiring 41 images for ~5nm wide spectral passbands between 450 and 950nm and investigate how the acquired hyperspectral data cube can be optimally analyzed for the tracking of skin features. Actually, contrary to what the title of the manuscript suggests, this article is all about data processing algorithms to compare the huge data sets and provide tracking of skin features. My main concern with this manuscript is that it is insufficiently detailed for another scientist to replicate the data set due to missing information. I would recommend revising following items before proceeding with publication:
- The abstract mentions “capturing image information beyond the visible spectrum is a novel technique…”. This is incorrect: this has been done for at least a few decades. What is new on this manuscript is the type of camera used. Please phrase this properly.
- The authors start abstract, introduction, conclusion talking about spine surgery. Hence it would be very welcome to the uninformed about this topic to present the problem in Figure 1 of the manuscript. How do the authors actually link the detection of skin features to the location of the spine?
- Furthermore, I would create a figure 2 based on the present figure 3 to explain the experimental set-up. Here, the main body of the manuscript lacks information for other researcher to replicate this set-up:
- Distance information missing
- Illumination source details missing. The type of illumination source must have a massive impact on this study. How do the authors guarantee sufficient power spectrum in the NIR?
- Lens information: actually, imaging with a lens in such a broad spectral range must be very challenging. Do the authors not struggle with focus mismatches in different wavelength bands?
- Camera description: “the hyperspectral camera is based on a tiled-filter approach. A figure describing this would be welcome? Why the limitation of 450nm? What is the frame rate of this camera? Is it fast enough to deal with subject moving?
- The authors allude to the fact that the light wavelengths probe the skin at different wavelengths. That’s correct, going further to the NIR allows one to probe deeper in the skin, provided that the power levels of the illumination source remain constant. If they don’t, the experimental recording system will probe not deeper. Actually, this topic of light penetration in skin is very complex and a lot of computational modelling has been done to investigate this, e.g. the work of Steven Jacques. Can the authors estimate the penetration depth of their system?
Reviewer 2 Report
This paper presents a study to evaluate the feasibility of hyperspectral imaging as a novel technique for detecting skin features to track patient position in navigated spinal surgery. Two local feature detection methods based on a conventional handcrafted local feature and a deep learning approach were employed and compared with the use of synthetic RGB images generated from the hyperspectral data. The proposed methodology based on the deep learning framework reveals the potential of HSI for tracking skin features, improving the results obtained with RGB data.
This reviewer considers the work is quite interesting for the scientific community and should be published after addressing some minor point that could improve the readability of the manuscript:
- Please, rephrase the sentence on line 15 and 16 of the abstract: “Capturing image information beyond the visible spectrum is a novel technique and amplifies the 16 collected data and provides information that cannot be obtained by the human visual system”.
- The term “Hyperspectral imaging” appears sometimes with the acronym (HSI) and other with the complete noun. I suggest to employ (after definition) the acronym HSI.
- Also, I suggest employing the acronym “HS” for the word “hyperspectral”.
- In the case of the MSI acronym that means “multispectral Imaging”, sometimes appears when referring to the images “MSI images”. In this case it should be “MS images” or “multispectral images”.
- In the introduction section, the paragraphs from line 76 to 92 (“A hyperspectral camera… …hyperspectral data cubes.”) should be moved to Materials and Methods section.
- Please, reorganize the position of the figures. In order to improve readability, figures/tables should appear relatively near after being cited in the text the first time. For this journal it is not necessary that figures/tables appear at the beginning of the page.
- In line 123, please, if possible, specify the brand of the illumination system and the power of the halogen light.
- Figure 4.a is not cited in the text, only Figure 4.b. Please, cite Figure 4.a.
- In line 147, please, if possible, specify the brand of the white reference plate and the percentage of reflection capable of providing.
- In line 234, please, include a reference to the Haar wavelet method.
- Maybe Figure 5 could be cited ad explained before in the text (for example in the last paragraph of page 7).
- In line 279, the potential of HSI respect to RGB is not only to penetrate deeper into tissue, but also to cover a wider spectrum.
- Section 3.2 is called “Band selection”, however a feature selection method (PCA) is employed for the experimentation. Maybe another name for this section could be suitable.
- Table 3 is cited before Table 2. Please, check the order of citations and the location of the figures/tables within the manuscript.
- In Table 2, for a better comparison among RGB and HSI, a summary of the best results obtained with HSI should be included. In addition, maybe subject 7, 8, 9, and 12 should be included and identified as n.a. (clarifying this in a footnote in the table).
- Figures 8 to 12 could be merge in a single figure where each independent figure can be identified as (a), (b), (c), …
Just a comment for future works, did you consider the use of high definition RGB imaging for tracking skin features and compare them with the use of HSI (that normally have lower spatial resolutions)?
I suggest accepting the manuscript after addressing these minor comments in the manuscript, since the proposed work it is highly valuable for the scientific community.
